# Characterization of Extracellular Vesicle-Coupled miRNA Profiles in Seminal Plasma of Boars with Divergent Semen Quality Status

**DOI:** 10.3390/ijms24043194

**Published:** 2023-02-06

**Authors:** Notsile H. Dlamini, Tina Nguyen, Ahmed Gad, Dawit Tesfaye, Shengfa F. Liao, Scott T. Willard, Peter L. Ryan, Jean M. Feugang

**Affiliations:** 1Department of Animal and Dairy Sciences, Mississippi State University, Mississippi State, MS 39762, USA; 2Department of Biological Sciences, Mississippi State University, Mississippi State, MS 39762, USA; 3Animal Reproduction and Biotechnology Laboratory (ARBL), Department of Biomedical Sciences, College of Veterinary Medicine and Biomedical Sciences, Colorado State University, Fort Collins, CO 80523, USA; 4Department of Pathology and Population Medicine, Mississippi State University, Mississippi State, MS 39762, USA

**Keywords:** seminal plasma, seminal quality, extracellular vesicles, miRNA, fertility, pig

## Abstract

Sperm heterogeneity creates challenges for successful artificial insemination. Seminal plasma (SP) surrounding sperm is an excellent source for detecting reliable non-invasive biomarkers of sperm quality. Here, we isolated microRNAs (miRNAs) from SP-derived extracellular vesicles (SP-EV) of boars with divergent sperm quality statuses. Raw semen from sexually mature boars was collected for eight weeks. Sperm motility and normal morphology were analyzed, and the sperm was classified as poor- or good-quality based on standard cutoffs of 70% for the parameters measured. SP-EVs were isolated by ultracentrifugation and confirmed by electron microscopy, dynamic light scattering, and Western immunoblotting. The SP-EVs were subjected to total exosome RNA isolation, miRNA sequencing, and bioinformatics analysis. The isolated SP-EVs were round spherical structures approximately 30–400 nm in diameter expressing specific molecular markers. miRNAs were detected in both poor- (*n* = 281) and good (*n* = 271)-quality sperm, with fifteen being differentially expressed. Only three (ssc-miR-205, ssc-miR-493-5p, and ssc-miR-378b-3p) allowed gene targeting associated with cellular localization (nuclear and cytosol) and molecular functions (acetylation, Ubl conjugation, and protein kinase binding), potentially impairing sperm quality. PTEN and YWHAZ emerged as essential proteins for protein kinase binding. We conclude that SP-EV-derived miRNAs reflect boar sperm quality to enable therapeutic strategies to improve fertility.

## 1. Introduction

Artificial insemination (AI) is the principal breeding technique used in most intensive pig production systems [1]. Semen is routinely harvested from sexually mature boars for breeding using extended-chilled semen, whose quality can drastically affect productivity (e.g., semen dose and fertility rate). Despite the maintenance of sires’ improved housing, the reproductive efficiency of boars still depends on many extrinsic and intrinsic factors, such as environment (or seasonal change), management skills, animal health, nutrition, age, and breed, affecting semen quality [1,2]. Nonetheless, the lack of reliable semen quality and fertility predictors drastically limits production in hog operations.

Most AI centers select boar ejaculates based on their genetic merit and sperm parameters, such as motility and morphology of high value (≥70%), evaluated using computer-assisted sperm analysis [3]. As a result, semen that does not meet this pre-screening is discarded, a process that may be costly during hot seasons due to high rejection rates [1]. Numerous observations indicate that not all selected ejaculates perform well [4]. Consequently, alternative studies have explored seminal plasma, the liquid non-cellular fraction of semen, as a promising source for non-invasive molecular characterization of boar semen. Seminal plasma (SP) is a heterogeneous mixture of secretions from the testes, epididymis, vas deferens, and accessory sexual glands comprising a wide array of bioactive molecules such as lipids, proteins, oligosaccharides, and minerals (calcium, magnesium, potassium, sodium, and zinc) and playing a regulatory role in sperm characteristics such as motility and morphology [5]. Like many biological fluids, SP is enriched with proteins and RNA that are encapsulated in extracellular vesicles (EVs) classified as either exosomes (40–120 nm diameter) or microvesicles (50–500 nm diameter) [5,6]. 

Seminal-plasma-derived EVs (SP-EVs) are produced by the epididymis and the prostate, characterized by high cholesterol and sphingomyelin [7]. These membranous vesicles have been isolated from human, rat, rabbit, ram, and bull seminal plasma [8,9,10,11,12]. Their presence as exosomes in boar seminal plasma (SP-EVs) is involved in capacitation-dependent cholesterol efflux and in maintaining sperm function during in vitro preservation [13]. In addition, EVs transport biomolecules such as proteins and RNA molecules involved in cell-to-cell communication in the male reproductive system [6]. SP-EVs contain microRNAs (miRNAs) that are single-stranded RNA molecules composed of endogenous 23-nucleotide RNAs regulating gene expression via post-transcriptional gene silencing in mammals [14]. MiRNAs are essential molecules capable of affecting boar sperm quality and, thus, influencing different biological processes such as sperm maturation and capacitation [15]. Thus, spermatozoa- and seminal-plasma-derived miRNAs can be potential biomarkers of sperm quality to use as a screening tool in the swine industry. 

The knowledge of the impacts of EVs-derived miRNAs on spermatozoa are still unsatisfactory, and the seminal plasma is an ideal milieu for a non-invasive search for biomarkers of good- or poor-quality semen. Here, we isolated EVs of sexually mature boar seminal plasma to profile miRNAs in relation to semen acceptance (good-quality) or rejection (poor-quality) for artificial inseminations. 

## 2. Results

### 2.1. Sperm Analyses of Good-Quality and Poor-Quality Semen 

Table 1 represents the sperm characteristics between good-quality and poor-quality semen. The percentages of total sperm motility and morphology in good-quality semen (87.9 ± 1.0; 84.4 ± 1.2) were significantly greater (*p* < 0.001) in comparison with poor-quality semen (56.4 ± 4.7; 49.9 ± 2.4). The percentage of progressive motility was higher in good-quality samples (*p* < 0.002). For abnormal morphology, only the proximal droplet significantly increased in the poor-quality sperm (*p* ≤ 0.019). There was no significant difference in the number of sperm abnormalities for distal droplet (*p* ≥ 0.156), coiled (*p* ≥ 0.071), and bent tail (*p* ≥ 0.057) between both semen groups. 

### 2.2. Characterization of Isolated Seminal-Plasma-Derived Extracellular Vesicles (SP-EVs)

Figure 1 shows the qualitative and quantitative characterizations of SP-EVs. Imaging with HR-TEM (Figure 1A) revealed the presence of EVs with round or cup-shaped morphology. Diameter size measurement with DLS (Figure 1B) indicated a symmetrical EVs size distribution ranging from 30 to 400 nm, corresponding to exosomes and microvesicles having similar size distributions in both SP-EV sample groups. Western immunoblotting (Figure 1C) indicated the presence of the specific molecular markers CD63 and CD81 at higher levels than CD9 in good-quality (EV-g) and poor-quality (EV-p) semen-derived SP-EVs. CD9 and CD63 are primarily present in microvesicles, while CD81 is found in exosomes. The cytoskeleton protein, ß-tubulin, was only detected in SP as an internal control. Altogether, the EVs separation by ultracentrifugation is within the recommended parameters validating EVs’ presence in boar SP and the absence of sperm cell contamination in EVs pellet.

### 2.3. Sequencing and Data Analysis

Quality control and sequencing—Averages of 1.66 ± 0.35 ng/µL and 2.04 ± 0.281 ng/µL total RNA were extracted from each poor- and good-quality SP-EV sample. RNA integrity was confirmed with the Bioanalyzer, indicating the absence of cellular RNA peak contaminations and rRNAs (18 s and 28 s). However, the presence of peaks around the small RNA size ranges was observed in all SP-EV samples. Figure 2 summarizes FastQC data following MultiQC (v1.7) analysis. The FastQC quality control tool for high throughput sequencing of samples indicated no significant differences in general statistics regarding the percentages of duplicate reads (94.5 ± 0.9% and 94.3 ± 0.7%) and GC (52.2 ± 1.0% and 52.6 ± 0.5%; Figure 2A), and total sequence read counts (23.4 ± 1.8 and 23.9 ± 2.3 million single-end reads; Figure 2B) between poor- and good-quality semen SP-EV samples, respectively. The average sequence length was 51 bp across all samples, and all samples exhibited high quality scores (Figure 2C).

Small RNA analysis and Principal component analysis—Read mapping to endogenous genome and annotated transcriptome indicated the proportions of 1.0% of total reads found without adapter and 9.4% unaligned to the genome. A proportion of 40.1% total reads aligned to the genome permitted the identification of various RNA biotypes (i.e., tRNA, miRNA, ncRNA, and piRNA). The principal component analysis (PCA) output and heat map clustering using the top 50 variable miRNAs indicated a clear separation between the two SP-EV groups (Figure 3A and Figure 3B, respectively). Two main components (PC1 and PC2) of samples explained about 53% of total variations between both semen quality groups.

MiRNA analysis and differential expression—An average of 79% of reads were mapped to the Sus scrofa genome, with an average of 30% of mapped reads annotated to miRNAs from the mirBase database. Totals of 271 and 281 unique miRNAs were detected with a minimum of five raw reads or counts (CPM) and were commonly shared between each replicate (n = 5) of good- and poor-quality SP-EV groups, respectively (Figure 4A). A total of 259 miRNA were shared between both experimental groups (Figure 4B), representing about 88% (259/293) of the total detected unique miRNAs. Proportions of 7.5% (22/293) and 4.1% (12/293) miRNAs were specific to poor-quality and good-quality sample groups, respectively. Pairwise comparative analysis revealed a total of 15 differentially detected miRNAs between poor-quality and good-quality sample groups (*p* < 0.05 and FDR < 0.1; Table 2). The volcano plot in Figure 4C illustrates the differential expression profile of significantly upregulated (n = 7) and downregulated (n = 8) miRNAs in the direction of the good-quality group. All differentially expressed miRNAs are listed in Table 2. None of the differentially expressed miRNAs were found to be amongst the most abundant (top ten) of the two groups (Figure 5), with the ssc-miR-10b, ssc-miR-10a-5p, ssc-miR-200b, ssc-let-7a, ssc-miR-191, ssc-miR-125b, and ssc-let-7f-5p found in both groups.

Functional annotation of miRNA targets—Only three differentially expressed miRNAs (ssc-miR-205, ssc-miR-493-5p, and ssc-miR-378b-3p) provided successful target gene prediction of 109 mRNAs. Functional annotation clustering indicated the highest and significant enrichment scores (>2.9x) with target mRNA transcripts associated with nuclear (nucleoplasm and nucleus) and cytosol cellular components. Additionally, the molecular function associated with protein kinase binding Gene Ontology term (GO:0019901) and functional annotation chart pathways related to post-translational modification through acetylation and Ubl conjugation were significantly enriched. Focusing on the unique significantly enriched pathway within the molecular function GO name (4.2x and *p*-value and FDR = 0.038), the protein kinase binding GO term contained 12 proteins, of which 11 were under the miRNA-205 (6) and miRNA-378b-3p (5) homologs. Figure 6 indicates the protein-to-protein interaction (PPI) network generated from gene products (proteins) associated with the “protein kinase binding” GO term. The PPI analysis confirmed the significant association of differential profiles of good and poor sperm quality SP-EV-derived miRNAs with the “protein kinase binding” GO term. All the studied proteins (10) revealed significant interactions (*p* = 0.01 and FDR = 0.02), with the PTEN and YWHAZ proteins appearing as the critical nodes with the most connections (eight and five, respectively).

## 3. Discussion

The paternal epigenetic inheritance shaping the offspring phenotype is likely influenced by environmental factors affecting sperm epigenetics, the main regulators in paternal effects [16]. During production and maturation, sperm cells are surrounded by seminal plasma, a complex mixture of macromolecules such as proteins, metabolites [17], and nucleic acids, including various small RNA biotypes [18]. It is expected that the production of such macromolecules in the vicinity of maturing spermatozoa would make them potential regulators of sperm function with potential effects on subsequent generation(s), but the full supportive molecular mechanisms are still unfolded. 

The present study focused on semen with spermatozoa showing poor or good motility and morphology parameters to allow molecular characterization of the corresponding seminal plasma. Indeed, mammalian spermatozoa undergo morphological and molecular changes during epididymal transit that affect sperm quality (motility and morphology), leading to the acquisition of fertilizing potential [19]. In boar farms, the standard threshold of 70% sperm motility and morphology is often set for semen retention for artificial inseminations. On the other hand, the semen of boars whose performance is mediocre, generally well below the threshold, is rejected (failed or poor-quality samples) [3]. The rejection rate could be exacerbated during critical or hot seasons, leading to high economic losses [1]. These poor-performing boars have significantly lower total sperm motility and higher abnormalities than their good counterparts. Sperm motility and morphology were higher in the good-quality semen, and these findings are highly correlated with the fertilizing potential of sperm. Boar sperm have been reported to lose cytoplasmic droplets during ejaculation, and these increase with collection frequency [20]. In the current study, sperm abnormalities were characterized by significantly higher proportions of sperm with proximal droplets in poor-quality semen. Droplets are cytoplasmic remnants in the neck region of the spermatozoon, indicating incomplete spermatogenesis. Proximal droplets could therefore reflect insufficient maturation. As a result, sperm quality is affected by low motility and morphology [21]. 

For molecular characterization, EVs are isolated through various techniques, including ultracentrifugation, density gradient, and size exclusion chromatography. There is no gold standard for EVs isolation. However, the selected approach depends on the characteristics of the starting material [22]. Here, we isolated EVs using ultracentrifugation, the most widely used method, because it allows large input volumes without additional chemicals [23]. Seminal-plasma-derived EVs have been detected and isolated in the seminal plasma of species such as humans, bovines, and boars [13,24,25,26]. The validation of isolated boar SP-EVs by TEM showed a heterogenous population of vesicle structures (microvesicles and exosomes). Isolated boar SP-EVs had a round or cup-shaped structure and were surrounded by a single or double membrane. The authors of [27,28] reported similar findings of cup-shaped or round morphology of boar SP-EVs, which is in line with the general description of EVs. Furthermore, DLS analysis showed that boar SP-EVs had a diameter of 30 to 400 nm, averaging 206 ± 26.3 nm, the typical size of EVs [5,6]. These findings also corroborate seminal plasma EVs derived from the prostate (i.e., prostasomes) with sizes of between 40 and 500 nm [29]. Although the size of exosomes and microvesicles are reported between 30–150 nm and 50–1000 nm, respectively [30], our isolation method provided about 69% of exosomes with a size range of 30 to 150 nm and 31% microvesicles with a range of 151 to 500 nm. Due to overlapping sizes of SP-EVs and similar morphology, other studies have reported different sizes of exosomes (50–150 nm or 30–120 nm) produced by the testes (30–200 nm), epididymis (50–250 nm), prostate (30–500 nm) [15,31], and microvesicles with a larger diameter (120–1000 nm) [31]. Indeed, techniques such as size exclusion chromatography and ultracentrifugation followed by filtration allow better separation of exosomes. However, the tissue specificity of their secretion (versus microvesicles) in sperm, triggering differential biological effects, is still ambiguous. We chose the ultracentrifugation protocol alone to ensure the collection of most EVs (exosomes and microvesicles) in boar semen for their putative biological roles. Western blot analysis showed that boar SP-EVs expressed higher levels of EVs specific markers CD9 and CD81 than CD63. These findings are in line with those reported by [13,28] in boar SP-EVs, indicating CD81/CD63 and CD9/CD63 as exosome and microsome markers, respectively. In addition to Bioanalyzer data, the absence of β-tubulin confirmed the absence of cytoplasmic contaminants. 

SP-EVs consist of miRNA cargo which reflects the status of the parent cell and can penetrate the sperm membrane in boars [13]. MiRNAs have been detected in SP-EVs of humans, bovines, and boars [28,32,33]. They play an essential role in intercellular communication between parent cells and their surrounding environment through RNAs, thus influencing sperm functions such as sperm motility. In the present study, deep sequencing of SP-EVs revealed the presence of various biotypes of small RNA (e.g., miRNA, piRNA, and tRNA). Totals of 271 and 281 miRNAs were detected in boar seminal plasma of good-quality and poor-quality groups, respectively. These findings align with a previous study reporting 288 known miRNAs in boar SP-EVs [28]. Similarly, 353 miRNAs were detected in SP-EVs of heat-stressed bulls [34]. 

Amongst the topmost abundant miRNAs, seven were shared across datasets (sc-miR-10b, ssc-miR-10a-5p, ssc-miR-200b, ssc-let-7a, ssc-miR-191, ssc-let-7f-5p, and ssc-miR-12b) and were expected to play critical regulatory roles in various physiological processes in sperm. For instance, ssc-miR-10a-5p regulates spermatogonial proliferation and differentiation, and it is highly expressed in porcine spermatogonia and spermatozoa [35]. Moreover, ssc-miR-200b targets the porcine spermatogenesis-associated serine-rich 2-like (SPATS2L) gene, significantly affecting litter size [36]. Like previous findings in humans and mice, ssc-miR-125b was highly expressed in porcine spermatogonia, where it can target LRRC8A, which activates the AKT pathway, which plays a key role in the survival of T cells. Together, ssc-miR-125b-LRRC8A-AKT can regulate the survival of porcine spermatogonia [37]. Furthermore, the let-7 family modulates IL-6, IL-10, and IL-13, which are associated with inflammatory responses [38]. The abundance of a let-7f-5p homolog in our boar SP-EV datasets may indicate immune functions as previously reported with let-7b in human SP-EVs [32]. Altogether, boar SP-EVs may have a modulatory role in sperm function and embryo development. 

A total of 15 miRNAs were significantly differentially expressed between the two SP-EV groups. Among the DE-miRNAs, only three miRNAs (ssc-miR-205, ssc-miR-493-5p, and ssc-miR-378b-3p) were revealed successfully for consideration of biological processes through target prediction. With a total target gene of 109 mRNA, pathway analyses indicated associations with various gene ontology (nuclear and cytosol localization and protein kinase binding) and post-translational modifications (acetylation and Ubl conjugation). These functional activities can potentially participate in the physical activity of seminal plasma proteins during phosphorylation and alter sperm function [39]. Acetylation (of Lys40 in αTubulin) is required for normal sperm flagellar function, including morphology and motility [39]. The ubiquitin system has been demonstrated to have a central role in spermatogenesis. Ubiquitin enzymes such as ubiquitin-conjugating enzymes (E2) mediate chromatin condensation, which involves the removal of the histone from the nucleosome, which is replaced by arginine-rich proteins called protamines [40]. Protamines bind to negatively charged DNA, resulting in tighter chromatin packaging in the sperm nucleus. Therefore, ubiquitination conjugation might play a regulatory role in the function of membranous organelles in male germ cells, including acrosome formation and nuclear condensation [41]. 

In this study, the use of PPI analysis, one of the major fields of systems biology [42,43], solidified the “protein kinase binding” GO term as a credible target of differential miRNA profiles in boar semen of divergent quality. The two miRNAs involved, miR-205 and miR-378b-3p, were downregulated in the good-quality group (−1.42x and −0.96x, respectively). The generated network indicated interconnections between all molecules, with central roles played by two strongly connected proteins, PTEN and YWHAZ, exhibiting the highest number of connections. From our datasets, these two gene products are under the control of miR-250 and miR-378, respectively, indicating a likely synergistic effect of both SP-EVs-derived miRNAs influencing sperm quality.

## 4. Materials and Methods

### 4.1. Semen Collection and Sample Preparation

Sperm-rich ejaculates (n = 64) were collected from forty healthy, sexually mature Duroc boars aged 1.5 to 2 years old (Prestage Farms Inc., West Point, MS, USA) maintained under controlled-environmental conditions of 20 to 22 °C with 59% relative humidity. Samples were collected using the gloved-hand technique for eight (8) weeks. All raw samples were chill transported in a Styrofoam shipping container to the laboratory within one hour for analyses. 

### 4.2. Sperm Motion and Morphology Analyses

Sperm aliquots were extended in phosphate-buffered solution (PBS) and incubated at 37 °C for 15 min and loaded (2 µL) into pre-warmed caffeine-free microscope chamber slides (Standard Count 4-chamber Slide Leja^®^, 20 microns, Nieuw Vennep, The Netherlands). Pre-set values of CASA were used (e.g., 60 frames/s; VAP and STR of progressive cells: 45 μm/sec and 45%, respectively; VAP and VSL cut-offs of slows cells at 20 and 5 μm/sec, respectively; magnification: 1.89x, and 37 °C), as previously described by [44]. Three fields per chamber-slide were considered to analyze the motility characteristics of spermatozoa using the CEROS II Computer-Assisted Sperm Analyzer, or CASA (IMV Technologies; Maple Grove, MN, USA). Each sample aliquot was run in triplicate (3 fields/chamber) with approximately 300 ± 3 (SEM) spermatozoa analyzed per chamber. The sperm motility (percent of the total, progressive, and rapid (≥30 μm/s), velocity parameters (μm/s; average path or VAP, straight line or VSL, and curvilinear or VCL), and other parameters such as the amplitude of lateral head aptitude (ALH, in μm), beat cross frequency (BCF, in Hz), straightness (STR; VSL/VAP × 100), and linearity (LIN; VSL/VCL × 100) ratios were also recorded. The proportions of morphologically abnormal spermatozoa, such as bent tail (BT), coiled tail (CT), distal (DD), and proximal (PD) cytoplasmic droplets were also recorded using CEROS II. From these analyses, semen samples that exhibited over 70% total motility and normal morphology were retained for AI. Thirty-three ejaculates were categorized as “good-quality” semen (>70%) and thirty-one ejaculates as “poor-quality” semen (<70%).

### 4.3. Isolation of Boar Seminal Plasma EVs 

Immediately after CASA analysis, semen samples (10 mL) were centrifuged at 800× *g* for 20 min at room temperature to collect spermatozoa. Corresponding supernatants were subjected to additional centrifugation (2000× *g* for 20 min) at 4 °C. The resulting supernatants (seminal plasma), devoid of cell debris and residual sperm cells, were collected and stored at −80 °C until EVs isolation. Frozen-thawed seminal plasma was centrifuged at 16,000× *g* for one hour at 4 °C. The resulting supernatants were ultracentrifuged (120,000× *g* for 70 min) at 4 °C. Visible EV pellets were rinsed twice with a high-purified cold PBS (5 mL) by ultracentrifugation (120,000× *g* for 70 min) at 4 °C. Cleaned EV pellets were resuspended in aliquots of 100 µL cold PBS and stored at −80 °C for further analysis. 

### 4.4. Characterization of SP Extracellular Vesicles 

Frozen-thawed seminal plasma EVs (SP-EVs) were characterized using various technical and analytical approaches: 

#### 4.4.1. Transmission Electron Microscopy 

Five μL of isolated SP- EVs were loaded onto formvar carbon-coated electron microscopy grids and incubated for 20 min at room temperature in a dry environment. They were then stained with 2% uranyl acetate (aqueous) for 5 min and embedded in a mixture of 4% uranyl-acetate and 2% methyl cellulose for 10 min on ice before air drying. EVs were examined and visualized using high-resolution transmission electron microscopy TEM-JEOL 2100 (JEOL Ltd., Tokyo, Japan) at 200kV TEM at the Institute for Imaging and Analytical Technology (I2AT, Mississippi State University). 

#### 4.4.2. Size Distribution 

Isolated SP-EVs were extended (20x) with high-purified PBS before dynamic light scattering (DLS) measurements. Analysis was performed according to the procedure described by [27]. Briefly, 1 mL of each sample was added to a polystyrene cuvette with a 10 mm pathlength and analyzed using a Nano Zeta Potential Analyzer (Brookhaven Instruments, Holtsville, NY, USA) at a wavelength of 659 nm with an angle of 90° at room temperature. Light scattering was recorded for 150 s, with three replicate measurements for each sample. DLS signal intensity was transformed to size distribution using Particle Solutions v.2.2 (Brookhaven Instruments, Holtsville, NY, USA) at the Dave C. Swalm School of Chemical Engineering, Mississippi State University. 

#### 4.4.3. Protein Extraction and Western Blotting 

Isolated SP-EVs were examined for specific markers using Western immunoblotting described by [45,46]. Total protein was extracted from three different isolated SP-EVs from “good” and “poor” quality semen using the complete RIPA lysis buffer supplemented with 1% protease inhibitor cocktail (Cat. No. 89900). After protein quantification (Cat. No. 23225; Rapid Gold BCA protein assay kit), extracted proteins of five boars were pooled according to semen quality group (n =  3 pools), and a total of 20 μg/pool was heat-denatured and loaded onto wells of SDS-PAGE (4–12.5% NuPAGE) gels for resolution at room temperature. In-gel proteins were transferred onto PVDF membranes, and the immunoblotting procedure was performed according to the anti-mouse WesternBreeze™ Chromogenic Detection kits (Cat. No. WB7105). Membranes were incubated for 60 min with primary antibodies (1/500 dilution), then washed and incubated for 30 min with the Alk-Phos conjugated anti-mouse secondary antibody. Immunodetected proteins were revealed by a chromogenic solution and protein bands were subsequently imaged for analysis and documentation. Unless otherwise indicated, all procedures took place at room temperature. Reagents were purchased from Thermo Fisher Scientific (Waltham, MA, USA). Primary antibodies (CD9: sc-13118, CD63: sc-5275, CD81: sc-166029, and β-tubulin: sc-58886) were purchased from Santa Cruz Biotechnology Inc. (Dallas, TX, USA). The anti β-tubulin served as a negative control.

### 4.5. Total RNA Isolation, Small RNA Sequencing, and Analysis

#### 4.5.1. Selected Samples 

In this experiment, SP-EVs of extreme motility and morphology parameters were selected as good (90.3 ± 2.2, n = 5) and poor (50.8–3.4, n = 5) samples. Extreme samples were selected from each quality group based on mean motility and morphology (±2SEM). 

#### 4.5.2. Total RNA Isolation 

Total RNA, including miRNA, was isolated using the Exosomal RNA Isolation Mini Kit (Norgen Biotek Corp., Thorold, Canada) according to the manufacturer’s instructions. RNA yield and quality were determined using a Thermo Scientific NanoDrop One spectrophotometer and Bioanalyzer, respectively. Samples were submitted to Norgen Biotek Corp. for small RNA sequencing using Illumina^®^ NextSeq 550 High Output Kit v2 platform and reagent, and sequencing libraries were prepared (Norgen Biotek Small RNA library prep Kit). 

#### 4.5.3. Small RNA-Sequencing Data Analysis 

Workflow was performed as previously described [47]. Clean data (clean reads) were processed from raw data and were mapped to the Ensembl Release 89 Sus scrofa genome (version 11.1) using RNAcentral. The quality of the raw FASTQ file was assessed using FASTQC version 0.11.4 (http://www.bioinformatics.babraham.ac.uk/projects/fastqc/, accessed on 29 July 2022). Raw sequencing reads were trimmed based on quality score (Q-score > 30) and read length (≥nucleotides). Sequence reads, aligned to the pig reference genome, were then used for annotation against pig precursor miRNAs and matured miRNAs in the miRbase database, release 20. The secondary structure was predicted using TargetScan and miRWalk. Normalization of raw expression data was conducted using the trimmed mean of M-values normalization method (TMM normalization) and presented as TMM-adjusted counts per million (CPM), which is based on log-fold and absolute gene-wise changes in expression levels between samples. 

#### 4.5.4. Differential Expression Analysis 

Filtration was set at a minimum of five CPM counts. Analyses were performed on the TMM normalized expression values using R version 3.6.3 and EdgeR statistical software package version 3.24.0. A miRNA with a log2 fold change of 1 ≥ log2 ≤ −1, *p*-value < 0.05 and an of average CPM > 5 was considered differentially expressed (DE). The potential pathways enriched by differentially expressed miRNAs were analyzed using DIANA-mirPath v3 (http://snf-515788.vm.okeanos.grnet.gr/, accessed on 4 November 2022). Hierarchical clustering analysis of all the differentially expressed miRNAs was performed using the ComplexHeatmap: 1.20.0. 

#### 4.5.5. MirRNA Target Gene Prediction and Bioinformatics Analyses 

The human miRNA homologous for the DE porcine miRNAs were identified from the miRbase database and subjected to mirRDB (release 5.0) and TargetScan Prediction tools to identify the target genes, which were used for functional annotations, enrichment, and pathway analyses through DAVID Bioinformatics Resources https://david.ncifcrf.gov/, accessed on 4 November 2022) [48,49]. Furthermore, corresponding proteins of selected transcripts were subjected to protein-to-protein interaction (PPI) analyses using STRING (https://string-db.org/cgi/input?sessionId = b4naasEUXe6L&input_page_show_search = on, accessed on 4 November 2022).

### 4.6. Statistical Analysis

Data analyses were performed using the Statistical Package for the Social Sciences (SPSS) for Windows, release 21.0 (SPSS Inc., IBM; Chicago, IL, USA). Data were analyzed with Student’s *t*-test. A *p*-value ≤ 0.05 was considered significant. All data were expressed as mean ± standard error mean (SEM). 

## 5. Conclusions

This study (1) examines the molecular characteristics of boar semen of poor-quality versus good-quality, (2) provides evidence for the presence of EV in the seminal plasma of boars with divergent semen quality, (3) provides profiling of plasma-derived EV seminal semen miRNAs, and (4) reveals differentially expressed miRNAs between the two semen groups, related to spermatogenesis and sperm function.

Further characterization (e.g., concentration and molecular content and roles) of EVs in boar semen could help develop non-invasive biomarkers of semen quality, enabling new therapeutic strategies for subfertility to improve the results of assisted reproduction in livestock.

## Figures and Tables

**Figure 1 ijms-24-03194-f001:**
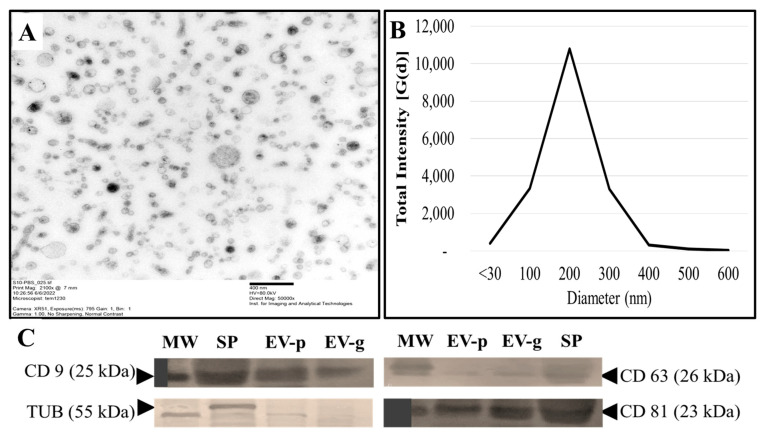
Characterization of extracellular vesicles isolated from boar seminal plasma (SP-EVs). The micrograph is a representative high-resolution transmission electron microscopy (HR-TEM) image of SP-EVs. Scale bar represents 400nm (**A**). Dynamic light scattering (DLS) size measurement and distribution of SP-EVs particles (**B**). Representative immunoblotting of SP-EV samples with exosome (CD63 and CD81) and microsomes (CD63 and CD9) markers (**C**).

**Figure 2 ijms-24-03194-f002:**
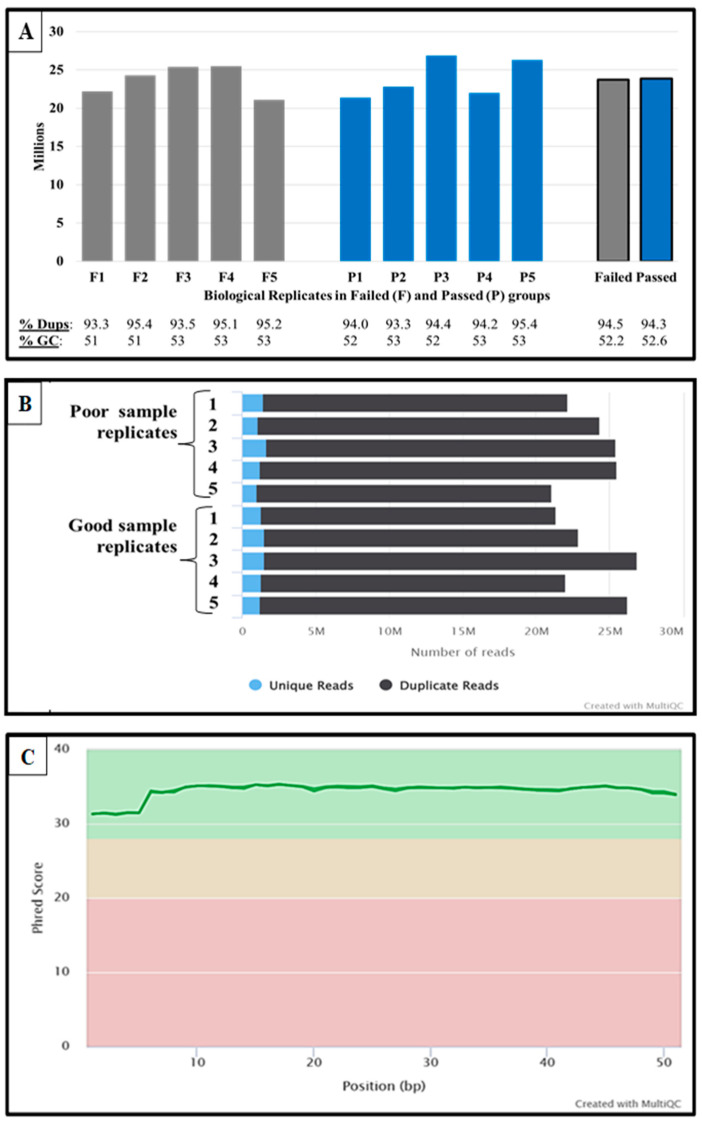
Deep-sequencing miRNA FastQC quality controls. FastQC total sequence reads, sequence counts, and mean quality scores are summarized in (**A**–**C**), respectively. Samples are five replicates of poor (failed) and good (passed) semen quality.

**Figure 3 ijms-24-03194-f003:**
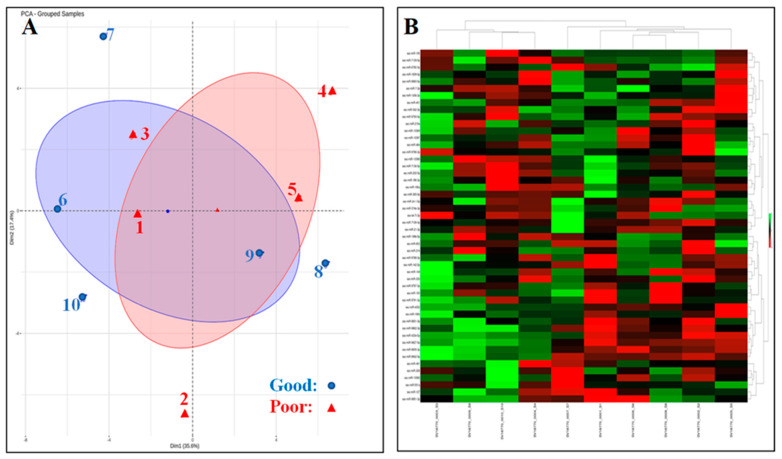
Principal component analysis (PCA) plot (**A**) and heat map with hierarchical clustering of miRNAs and samples (**B**). A total of 50 miRNAs having the highest %CV were used. The color code in the heat map indicates higher (green color) or lower (red color) miRNA expression relative to general mean expression. Five differentially expressed miRNA are found in this heat map (ssc-9828-3p, ssc-miR-7139-3p, ssc-miR-1277, ssc-miR-10386, ssc-miR-9788-3p).

**Figure 4 ijms-24-03194-f004:**
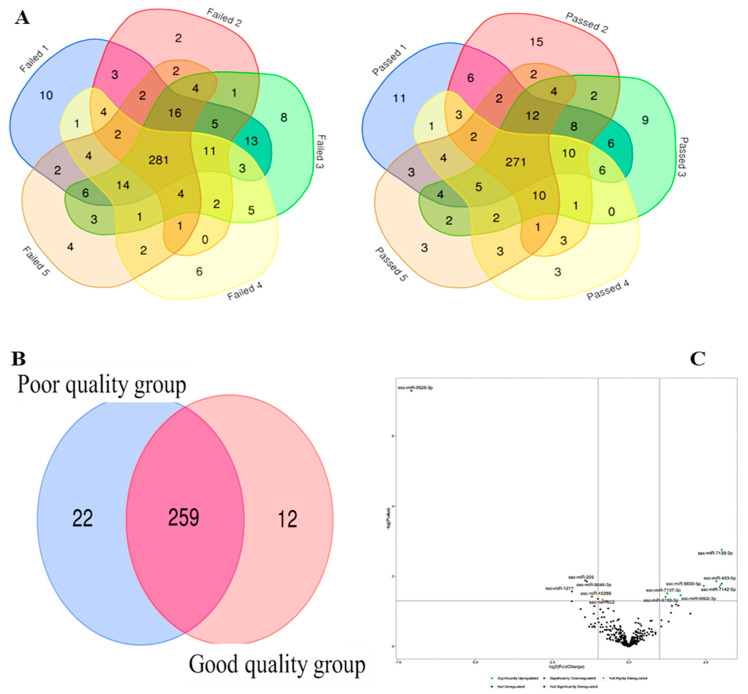
Total detected and differentially expressed miRNAs. Venn diagram represents the totals of 271 and 281 unique miRNAs detected in all replicates (n = 5) of each good (passed) and poor (failed) semen quality group (**A**). Venn diagram comparison of both groups reveals the presence of 88% shared miRNAs (**B**). Volcano plot illustrates differentially expressed miRNAs through fold change deregulation (≥2 or log2 fold change of ≥1 or ≤−1, with FDR≤0.05 or -log10 FDR of ≥1.30103) (**C**).

**Figure 5 ijms-24-03194-f005:**
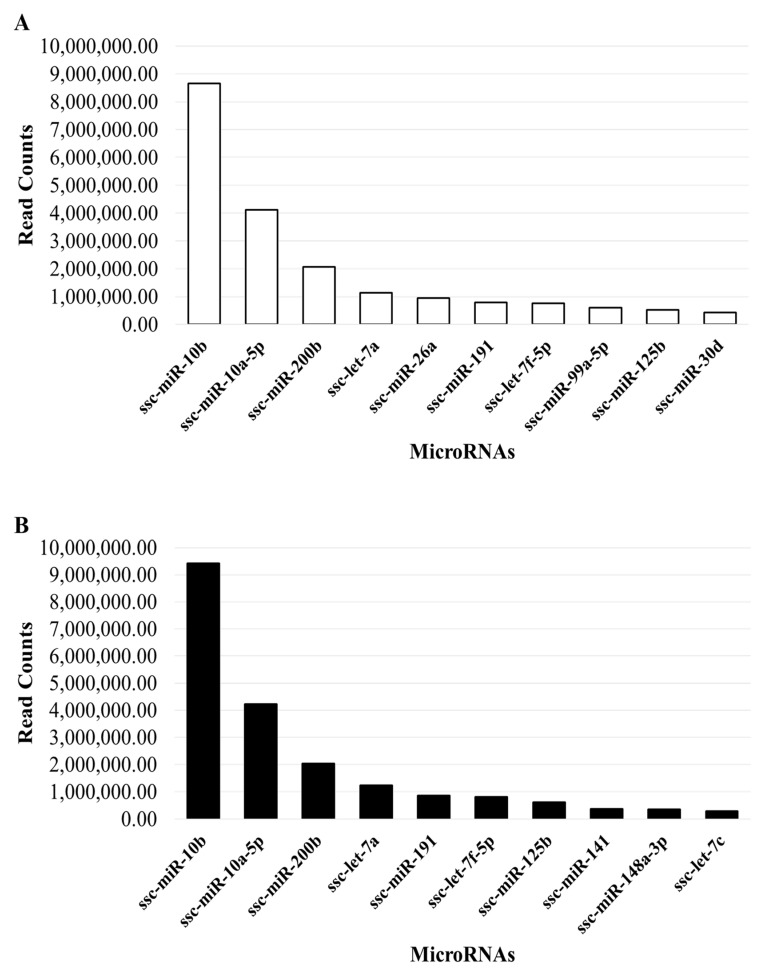
The top 10 most abundant miRNAs in SP-EVs of good (**A**) and poor (**B**) semen quality.

**Figure 6 ijms-24-03194-f006:**
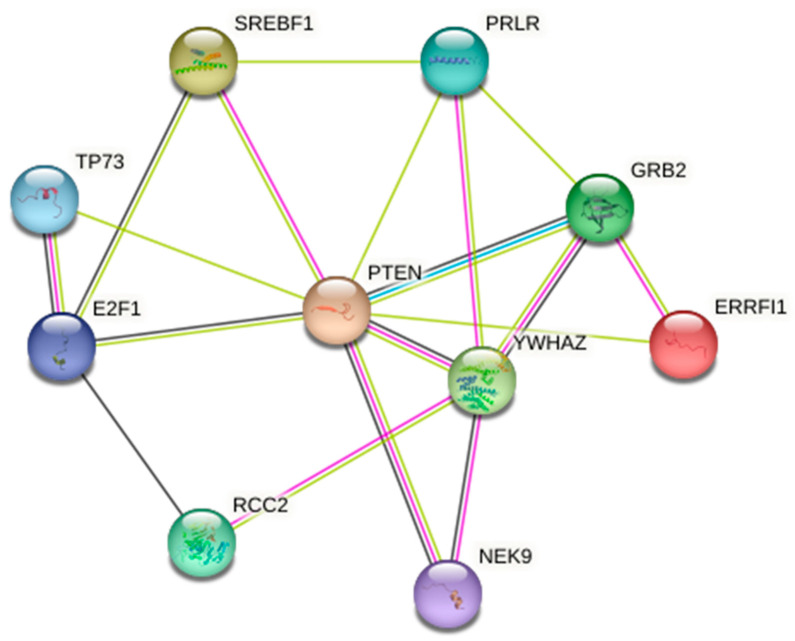
Protein-to-protein interaction network of proteins related to the “protein kinase binding” Gene Ontology (GO) term.

**Table 1 ijms-24-03194-t001:** Semen characteristics in good-quality and poor-quality boar semen.

Sperm Characteristics	Good-Quality Semen	Poor-Quality Semen	*p*-Value
Total motility (%)	87.9 ± 1.0 ^a^	56.4 ± 4.7 ^b^	<0.001
Progressive motility (%)	51.9 ± 3.4 ^a^	36.0 ± 4.4 ^b^	0.002
Normal morphology (%)	84.4 ± 1.2 ^a^	49.9 ± 2.4 ^b^	<0.001
Bent tail (%)	0.7 ± 0.2	1.8 ± 0.4	0.071
Coiled tail (%)	3.4 ± 0.5	5.3 ± 0.8	0.057
Distal droplet (%)	6.5 ± 0.7	8.1 ± 0.8	0.156
Proximal droplet (%)	16.5 ± 1.7 ^a^	24.6 ± 2.5 ^b^	0.019

Data are mean ± SEM; ^ab^ means without a common superscript letter in the same line differ significantly (*p* < 0.05).

**Table 2 ijms-24-03194-t002:** List of differentially expressed miRNAs in direction of good semen quality group.

miRNA Name	Log_2_ Fold Change	Log CPM	*p*-Value	FDR
ssc-miR-9828-3p	−7.07	−1.19	5.09 × 10^−8^	1.79 × 10^−5^
ssc-miR-7142-5p	3.02	0.58	1.6 × 10^−2^	9.4 × 10^−1^
ssc-miR-7139-3p	3.02	1.79	1.75 × 10^−3^	3.1 × 10^−1^
ssc-miR-493-5p	2.85	0.32	1.39 × 10^−2^	9.4 × 10^−1^
ssc-miR-9830-5p	2.43	0.80	1.86 × 10^−2^	9.4 × 10^−1^
ssc-miR-1277	−1.85	−0.79	2.72 × 10^−2^	1
ssc-miR-9862-3p	1.68	4.82	3.48 × 10^−2^	1
ssc-miR-205	−1.42	5.41	1.31 × 10^−2^	9.4 × 10^−1^
ssc-miR-9846-3p	−1.37	5.41	1.40 × 10^−2^	9.4 × 10^−1^
ssc-miR-7137-3p	1.25	5.94	3.08 × 10^−2^	1
ssc-miR-10386	−1.2	2.43	3.79 × 10^−2^	1
ssc-miR-9788-3p	1.19	1.23	3.88 × 10^−2^	l
ssc-miR-802	−1	2.09	4.40 × 10^−2^	1
ssc-miR-378b-3p	−0.96	2.40	3.07 × 10^−2^	1
ssc-miR-369	−0.86	2.48	4.56 × 10^−2^	1

Significant fold change expression is determined for −1 > Log fold change (Log2FC) > +1 with *p*-value < 0.05. CPM = counts per millions. FDR = false discovery rate.

## Data Availability

All data are provided in the manuscript.

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
