# Peer review of "Characterization of Extracellular Vesicle-Coupled miRNA Profiles in Seminal Plasma of Boars with Divergent Semen Quality Status"

_ijms, 2023, doi:10.3390/ijms24043194_

Round 1

Reviewer 1 Report

The research screened the miRNA profiles of the SP-derived extracellular vesicles with divergent boar semen quality status. They found that three (ssc-miR-205, ssc-miR-493-5p and ssc-miR-378b-3p) were differentially expressed in the extracellular vesicles  between good and poor quality semen. They concluded that the target mRNAs of the three mircoRNAs are associated with cellular localization and molecular functions, potentially impairing sperm quality. The topic is interesting. However, there are two major concerns.

1.      In the manuscript, they authors did not measure the three miRNAs in the  the SP-derived extracellular vesicles using RT-PCR. They have to perform the experiments to confirm the expression.

2.      The SP-derived extracellular vesicles are able to transfer miRNAs and the other molecules the sperm membranes. Thus, I strongly recommend the authors perform more experiments to detect the expression of the top three differentially expressed miRNAs in the sperm, and compare their expression between the high quality and poor quality samples.

In addition, some minor concerns.

1.  How many boars were used in the experiments. They authors need add the information.

2. Fig 1C. The quality is very poor. The authors should provide good images.

3.  Fig 2 and Fig 4B are too big. They should be adjusted.

4. I think the Discussion needs to be reduced.

Author Response

Response to reviewer #1’ concerns

Comments and Suggestions for Authors

The research screened the miRNA profiles of the SP-derived extracellular vesicles with divergent boar semen quality status. They found that three (ssc-miR-205, ssc-miR-493-5p and ssc-miR-378b-3p) were differentially expressed in the extracellular vesicles between good and poor quality semen. They concluded that the target mRNAs of the three mircoRNAs are associated with cellular localization and molecular functions, potentially impairing sperm quality. The topic is interesting. However, there are two major concerns.

  1. In the manuscript, they authors did not measure the three miRNAs in the the SP-derived extracellular vesicles using RT-PCR. They have to perform the experiments to confirm the expression.

Response: The authors understand the concern of the reviewer. Similar studies that investigated the presence of miRNA for potential biological effects were limited to identifying predicted mRNA targets. The current study aimed at identifying those miRNAs.  However, we will conduct miRNA detection/validation (qPCR) in future studies using SP-EVs and spermatozoa. This detection will be combined with further investigation of derived gene expression products that may influence sperm viability and fertilizing potential.

  1. The SP-derived extracellular vesicles are able to transfer miRNAs and the other molecules the sperm membranes. Thus, I strongly recommend the authors perform more experiments to detect the expression of the top three differentially expressed miRNAs in the sperm, and compare their expression between the high quality and poor quality samples.

Response: The authors fully agree with the need for such a study.  As indicated in our answer above, we want to implement it as a part of a functional investigation of their effectiveness.

In addition, some minor concerns.

  1. How many boars were used in the experiments. They authors need add the information. Fixed on lines 323-324.

  1. Fig 1C. The quality is very poor. The authors should provide good images. New electrophoresis gels have been run to generate new images that were added. An updated Fig 1C is provided.

  1. Fig 2 and Fig 4B are too big. They should be adjusted. Figures have been sufficiently reduced.

  1. I think the Discussion needs to be reduced. The authors politely disagree with the reviewer as this section is balanced to address most of the results.

Reviewer 2 Report

This paper deals with a highly up-to-date topic since the search for novel markers to predict sperm quality has become an interesting strategy in animal andrology. The study is well designed and employs a wide array of methodologies to detect and characterize microRNAs isolated from seminal plasma extracellular vesicles that could be indicative of suitability of a semen sample for artificial insemination in swine production. The manuscript reads well, although I do have several suggestions to further clarify the study for potential readers of the journal:

-          Although the authors mention the sample number, the actual number of boars included in the experiments. Does each sample represent one boar or were they collected repeatedly from a certain number of animals? Also, how old were the animals? Were they similar in age? If not, this could have an impact on the seminal plasma composition as well as the resulting semen quality.

-          Whilst I understand that the Westerns were used as a secondary technique to the primary genetic screening of the vesicles, it would be useful to add more specifics to the protocol used in the study. What isotype/clonality/catalog number were the antibodies used for the detection? What blocking solution was used? How were the blots evaluated?

-          Although these were previously specified, abbreviations should be explained in the legend of each Table or Figure if used.

-          Out of all miRNAs detected and evaluated, is there any that may present with the highest potential or value to be used as a predictor of semen quality in the future?

-          I would appreciate if the authors could briefly discuss any limitations of the study, as well as to showcase how their findings could be translated into future studies or a practical application of miRNAs in the prediction of the suitability of a specific semen sample for artificial insemination. Finally, a take home message or highlights could strengthen the importance of the collected data.

Author Response

Response to reviewer #2’ concerns

Comments and Suggestions for Authors

This paper deals with a highly up-to-date topic since the search for novel markers to predict sperm quality has become an interesting strategy in animal andrology. The study is well designed and employs a wide array of methodologies to detect and characterize microRNAs isolated from seminal plasma extracellular vesicles that could be indicative of suitability of a semen sample for artificial insemination in swine production. The manuscript reads well, although I do have several suggestions to further clarify the study for potential readers of the journal:

- Although the authors mention the sample number, the actual number of boars is included in the experiments. Does each sample represent one boar or were they collected repeatedly from a certain number of animals? Also, how old were the animals? Were they similar in age? If not, this could have an impact on the seminal plasma composition as well as the resulting semen quality.

Response: The authors thank the reviewer for pinpointing these important details that were overlooked. Semen samples were obtained from 40 boars aged 1.5 to 2 years during eight weeks. Information can be found in lines 323-326.

- Whilst I understand that the Westerns were used as a secondary technique to the primary genetic screening of the vesicles, it would be useful to add more specifics to the protocol used in the study. What isotype/clonality/catalog number were the antibodies used for the detection? What blocking solution was used? How were the blots evaluated?

Response: All requested information is added in lines 381 to 397. The WIB examined the presence or absence of proteins. We did not quantify the signal band intensities.

- Although these were previously specified, abbreviations should be explained in the legend of each Table or Figure if used.  Fixed wherever needed in tables and figures.

- Out of all miRNAs detected and evaluated, is there any that may present with the highest potential or value to be used as a predictor of semen quality in the future?

Response: The study identified a set of 15 miRNA that were differentially detected between good and poor semen that could serve as potential markers. Most interestingly, three of the differentially expressed miRNAs are fully annotated to enable further characterization. More details of the three miRNAs are given in lines 290-312.

- I would appreciate it if the authors could briefly discuss any limitations of the study, as well as showcase how their findings could be translated into future studies or a practical application of miRNAs in the prediction of the suitability of a specific semen sample for artificial insemination. Finally, a take-home message or highlights could strengthen the importance of the collected data.

Response: The authors thank the reviewer for requesting additional information that strengthens the manuscript.  Further information is provided in lines 263-267 to highlight potential limitations and rationale for using the proposed ultracentrifugation protocol. The Conclusion has been revised for clarity on the take-home messages (Lines 449-456).

Round 2

Reviewer 1 Report

     The research is interesting and well designed. I generally agree with the author's responsebut I still suggest adding the q-PCR validation of miRNAs in the SP-derived extracellular vesiclesIn particular the three miRNAs (ssc-miR-205, ssc-miR-493-5p and ssc-miR-378b-3p).